# Working Conditions as Risk Factors for Depressive Symptoms among Spanish-Speaking Au Pairs Living in Germany—Longitudinal Study

**DOI:** 10.3390/ijerph18136940

**Published:** 2021-06-28

**Authors:** Bernarda Espinoza-Castro, Tobias Weinmann, Rossana Mendoza López, Katja Radon

**Affiliations:** 1CIH-LMU Center for International Health, LMU University Hospital Munich, 80336 Munich, Germany; katja.radon@med.uni-muenchen.de; 2Institute and Clinic for Occupational, Social and Environmental Medicine, LMU University Hospital Munich, 80336 Munich, Germany; Tobias.Weinmann@med.uni-muenchen.de; 3Center for Translational Research in Oncology, Instituto do Câncer do Estado de São Paulo, Sao Paulo 01246, Brazil; rossana@alumni.usp.br

**Keywords:** au pair, caregivers, migrants, working conditions, depressive symptoms, Latin American, Spanish-speaking migrants

## Abstract

Previous studies have shown poor working conditions and poor mental health among au pairs. However, there are limited longitudinal approaches to these conditions. Therefore, the main objectives of this study were to assess the occurrence of depressive symptoms longitudinally and to analyze the association between sociodemographic characteristics, working conditions and violence at work with depressive symptoms over time among Spanish-speaking au pairs living in Germany. A prospective cohort study was performed with three measurement intervals, which included 189 participants. Depressive symptoms were assessed by the Patient Health Questionnaire (PHQ-9). Generalized Estimating Equation (GEE) models were implemented to estimate the association between predictors and depressive symptoms. Au pairs who worked >40 h per week were more than three times more likely to experience depression than those who did not (OR: 3.47; 95% CI: 1.46–8.28). In addition, those exposed to physical violence were almost five times more likely to suffer from depression (OR: 4.95; 95% CI: 2.16–9.75), and au pairs who had bad schedule adaptation to social and family commitments had twice the risk of depression than those who did not (OR: 2.24; 95% CI: 0.95–5.28). This knowledge could be of interest for future au pairs, host families, au pair agencies and policy makers. Together, they could improve awareness and monitoring of au pair working conditions.

## 1. Introduction

In the 1950s, the number of housewives in Europe decreased due to industrialization, women’s participation in the labor market and the enrichment of the middle class in the global North, among other causes [1]. Therefore, the number of paid domestic helpers, mainly female migrant workers, increased, and new migration forms such as au pairs emerged [1]. An “au pair placement provides an opportunity for young people to learn a language and a culture abroad, while temporarily (1–2 years) living as a ‘member’ of a host family and providing light domestic work and childcare for this family” [2]. The term au pair comes from the French ‘at par,’ ‘at equal shares’ or ‘on mutual terms.‘ It refers to the mutual benefit gained from au pairs and host families [3].

During the last decade, the number of au pairs in Europe has been on the rise. On the one hand, middle class European families cannot afford paid domestic workers, there are limited public childcare centers and private centers are costly [4]. On the other hand, au pair programs are the cheapest way for young foreigners to achieve personal development, such as learning or improving their language skills, traveling independently or experiencing other cultures [5]. In Germany, from 2012 onwards, there has been an increase in the number of au pairs every year [6]. In 2018, a total of 14,000 au pairs started working in Germany, most of them coming from Colombia, Georgia, and Ukraine [7]. In the last three years, Germany perceived a marked growth of Latin American au pairs [7]. Consequently, Spanish-speaking and especially Latin American au pairs represent one of the largest groups in Germany (e.g., 9.2% Colombians and 4.3% Mexicans).

Latin American au pairs are usually highly educated, come from families with middle or high socio-economic status, are still dependent on their parents and their major desire is achieving personal development rather than maximizing their earnings [8]. Furthermore, some of them might perceive childcare as a low-skilled job, typically done by a domestic helper from a different economic or ethnic group [8]. This perception might turn into poor mental health as presented in Espinoza et al., who identified 45% prevalence of distress associated with working below skill level among Latin American migrants living in Germany [9]. In addition, some conflicts may appear due to cultural differences. High context (HC) cultures such as those in Spanish-speaking countries present indirect and implicit communication through gestures or body language, while low context (LC) cultures such as those in northern European countries have an explicit and direct communication style [10]. These differences could cause misunderstandings and communicational dissonance, especially during disputes, negotiations or conflicts [11].

Another challenge is that Spanish-speaking au pairs might experience poor working conditions. Firstly, residence permission in Germany for non-EU au pairs is tied to the host family. The au pairs’ immigration status excludes employee protection laws such as minimum wage (they receive only pocket money) or forty hours of work per week as a full-time employee [2]. Furthermore, the live-in structural dependency of the employer/host family without any direct outside supervision might contribute to poor working conditions, abuse or even harassment among au pairs [2]. For example, according to a recent survey, the main problems which au pairs face in Germany are working overtime and having unclear working instructions [6]. Another study from Germany reported a prevalence of twelve percent for violence and three percent for sexual abuse among Latin American au pairs [12].

From a theoretical perspective, those working conditions can be seen as factors that in occupational stress models such as the Job-Demands Resources (JD-R) model are typically characterized as job demands [13]. Via stress-related mechanisms, those demands can affect a worker’s well-being, and are thus a potential risk factor for individuals’ mental health. According to the JD-R model, those demands include physical, psychological, organizational and social aspects of the job; we assume that au pairs face demands in several of those categories (e.g., overtime and unclear working instructions as organizational aspects, harassment as a physical aspect, being separated from their families for the first time in their life as a psychological/emotional demand) [13].

Consequently, we hypothesize that au pairs might be at elevated risk of developing poor mental health and poor psychological well-being when childcare duties are more difficult than they expected, household tasks are added to their workload and cultural and foreign language challenges emerge [14]. For instance, in an earlier cross-sectional study, we observed a 19% prevalence of Major Depressive Syndrome (MDS) particularly among experienced Spanish-speaking au pairs living in Germany [15].

Thus, the length of stay in the host country is another factor that influences migrants’ mental health and well-being, and it is consistent with the “Healthy Migrant Effect.” This effect refers to two phenomena: (1) the healthiest population tends to migrate, and (2) migrants’ physical and mental health deteriorates or even disappears in a relatively short period of time [16].

To increase awareness among decision-makers, public health bodies and host families about potential mental health problems among au pairs and to potentially prevent depressive symptoms in au pairs, it is important to determine which factors are linked to depressive symptoms and when they emerge. Depressive symptoms are defined as “somatic and non-somatic factors that in sum determine the presence or absence of several subtypes of depression, including major depressive episodes” [17,18]. We focus on the Spanish-speaking population because of the challenges already mentioned, such as educational mismatch (they are typically over-educated), poor living and working conditions, violence at work and the conflict between HC and LC cultures.

Based on the above-mentioned theoretical considerations regarding job demands and their potential effect on mental health and well-being, and following-up on a previous cross-sectional analysis [15], we hypothesize that poor working conditions and violence at work are associated with depressive symptoms among Spanish-speaking au pairs, and that these symptoms increase with the time spent in the host country. Thus, we aimed to assess the occurrence of depressive symptoms with an explorative quantitative and longitudinal approach to analyze the association between sociodemographic characteristics, working conditions and violence at work with depressive symptoms over time.

## 2. Materials and Methods

### 2.1. Study Design and Population

We performed a prospective cohort study with three measurement intervals: (1)Baseline (T0): carried out when the au pair arrived in Germany (less or equal to three weeks from arrival).(2)1st follow-up (T1): one month after the initial assessment.(3)2nd follow-up (T2): six months after the initial assessment.

Inclusion criteria were: (1) being a “newcomer au pair”, i.e., living in Germany less or equal to three weeks at baseline, (2) being born in a Spanish-speaking country and (3) being aged between 18 and 28 years old (the age required in Germany to work as an au pair from non-EU countries).

### 2.2. Recruitment of Participants

The data was collected from August 2018 to April 2020. As no official registry of Spanish-speaking au pairs in Germany exists, we applied two snowball recruitment methods: 1) conventional snowballing and 2) snowballing via Facebook. In a previous article, we described these methods in greater detail [15]. For conventional snowballing, we contacted 23 au pair agencies in Germany, Spain and Latin America. We asked them to share the study invitation email and the link to the online survey with au pair candidates. For Facebook, we distributed the link for the online survey through paid advertising. Furthermore, we posted the online survey’s link in 58 Facebook groups of Spanish-speaking au pairs living in Germany.

For the follow-ups, we emailed the questionnaires to participants who attended the baseline study with two reminders to ensure a higher response. The first reminder was sent after two weeks and the second after four weeks from the initial follow-up. As an incentive, we provided an online shopping voucher worth five Euros to those participants who answered the entire questionnaire for the baseline study and an additional five Euro voucher for each follow-up.

In this way, we approached a total of 422 Spanish-speaking au pairs living in Germany. According to the inclusion criteria, 189 participants were eligible for the study. A total of 90 (47.6%) participants dropped out of the study during the follow-ups, so that for 99 individuals data was available for baseline and both follow-ups (Figure 1).

All participants gave informed consent to participate in the study after they read the objectives, procedures and ethical principles of the study. To maintain the participants’ anonymity, they created their own identification code at T0. This code consisted of three letters and three numbers (e.g., HGT968). Participants were able to resign from the study at any time. The study was approved by the Ethical Committee of the Medical Faculty at the Ludwig Maximilian University of Munich (project number 18–139).

### 2.3. Questionnaire Instruments

The questionnaire included 33 questions to assess depressive syndromes, working conditions, violence at the workplace and socio-demographic characteristics. The questions were provided in Spanish and were administered as an online questionnaire using LimeSurvey^®^.

Depressive Syndromes were evaluated by the Spanish version of the Patient Health Questionnaire depression module (PHQ-9) [19]. This tool is a 9-item Likert-type scale designed to assess depression during the previous two weeks. It follows nine depressive symptoms criteria from the DSM-IV: “(1) Depressive mood, (2) Loss of interest or pleasure in almost all activities, (3) Significant (more than 5% in a month) unintentional weight loss/gain or decrease/increase in appetite, (4) Sleep disturbance, (5) Psychomotor changes (agitation or retardation) severe enough to be observable by others, (6) Tiredness, fatigue, or low energy, or decreased efficiency with which routine tasks are completed, (7) A sense of worthlessness or guilt, (8) Difficulty thinking, concentrating, or making decisions, and (9) Recurrent thoughts of death or suicidal ideation, or suicide attempts” [20,21]. Each symptom is rated on a 4-point Likert scale ranging from 0= “not at all”, 1= “several days”, 2= “more than the half of the days”, to 3= “nearly every day” [20]. According to PHQ-9, subjects are classified in four categories: (1) Depression-symptom-screen negative (DS-): none of the positive responses are present “more than half of the days;” (2) Depression-symptom-screen positive (DS+): at least one of the positive responses are present at “more than half of the days excluding depressed mood and lack of interest;” (3) Other Depressive Syndrome (ODS): two to four positive responses are present in “more than half the days” (suicide item: “several days or more”) including at least depressed mood or lack of interest and (4) Major Depressive Syndrome (MDS) at least five positive responses are present in “more than half the days” (suicide item: several days or more) including depressed mood or lack of interest [20,22]. For our analysis, we defined the outcome variable “depressive symptoms” as a binary variable (yes or no), where “yes” included all the participants with DS+, ODS and MDS and “no” included participants classified as DS-.

Socio-demographic characteristics, working conditions and violence at the workplace were assessed by the Spanish short version of the European Working Conditions Survey [23] and the Quality of Life and Employment, Labor, and Health Conditions First National Survey (ENETS) [24]. Working conditions included the following variables: working hours per week (≤40 h or >40 h), working on holidays (yes or no), days off per week (one day or two days), work schedule’s adaptation to social and family commitments (good or bad), au pair agency contract (yes or no), extra hours of babysitting (yes or no, where babysitting means taking care of children while parents are away from home due to leisure activities) and additional jobs besides au pair (yes or no). Moreover, violence at work included: physical violence by the host children (yes or no), verbal offenses (yes or no) and violence at the workplace (yes or no, where violence at the workplace means physical violence or sexual harassment by the host family).

Finally, socio-demographic characteristics included sex (male, female), age (18–21, 22–24, 25–28 years), higher level of education (yes or no, where “yes” means at least one year at a higher education institution), region of origin (Spain, Colombia, Mexico and Central America, and South America without Colombia), region of residence in Germany (northern, southern, eastern, or western), settlement of residence in Germany (“towns” <100,000 inhabitants, “minor cities” 100,000–500,000 inhabitants, “major cities” >500,000 inhabitants) and follow up times (T0 = baseline measure; T1 = 1-month; T2 = 6-month follow up).

### 2.4. Statistical Analysis

Data analysis was performed with SPSS^®^ version 25.0 (IBM, Armonk, NY, USA). First, in the descriptive analysis, we compared participants who remained in the whole study, from T0 to T2, versus participants who dropped out during the follow-ups. Nominal and ordinal variables were described as absolute and relative frequencies. Secondly, for the bivariate analyses, we used Fisher’s exact test to assess the association between sociodemographic characteristics and working conditions with depressive symptoms. In order to assess the association between the dependent variable depressive symptoms in T0 and T1 with T2, we used McNemar’s test on paired data.

Finally, we implemented Generalized Estimating Equations (GEEs) to analyze the within-subject correlations throughout time [25]. This method is useful for longitudinal data, especially for discrete outcomes [26] and for “repeated measures using a common working correlation matrix for the longitudinal responses of each subject” [27]. Furthermore, GEEs include a semi-parametric regression-based strategy for handling missing completely random data in longitudinal studies due to dropouts during the follow-ups [25,28,29]. This strategy is commonly used when the probability density model for the measurement process is difficult to fully specify. Hence, GEEs work only with means and variances, and a common correlation matrix for the multivariate measurements instead of the distribution of the multivariate data [30]. The selected GEE adjusted model presented the lowest Akaike Information Criterion (AIC) and Bayesian Information Criterion (BIC).

## 3. Results

Most of the participants who remained in the study across all three time points were female (89.7%) and highly educated (86.0%). More than half of them were from Colombia (54.6%), followed by Mexico (22.7%). The prevalence of MDS was higher in participants who dropped out the study during follow-ups than in those who remained in the study (12.2% vs. 3.1%;). In addition, participants who dropped out tended to be younger (18–21 years) than those who remained in the study (47.3% vs. 30.9%). Regarding the remaining variables, there were no meaningful differences between the participants who were lost during the follow-ups and the participants who completed the study (Table 1).

The prevalence of depressive symptoms over time was 27.6% at baseline, 28.1% at the first follow-up and 26.5% at the second follow-up. In the bivariate results, working more than forty hours per week, working on holidays, bad schedule adaptation to social and family commitments, violence at work, physical violence by the host children and verbal offenses were associated with depressive symptoms (Table 2 and Table 3).

The adjusted GEE model showed an association between working more than forty hours per week, experiencing physical violence from the host children, and having bad schedule adaptation to social and family commitments with depressive symptoms. Au pairs who worked more than forty hours per week were about three times more likely to experience depression than those who did not (odds ratio [OR]: 3.47; 95% confidence interval [95%CI]: 1.46–8.28). In addition, those exposed to physical violence were almost five times more likely to suffer from depression than those who were not (OR: 4.95; 95% CI: 2.16–9.75). Finally, au pairs who had bad schedule adaptation to social and family commitments had twice the risk of depression than those who did not (OR 2.24; 95% CI: 0.95–5.28) (Table 4).

## 4. Discussion

This study investigated predictors of depressive symptoms among a cohort of 189 Spanish-speaking au pairs living in Germany along three follow-ups. The results showed a high prevalence of depressive symptoms among the au pairs included in the study. Moreover, the observed data support the hypothesis of an association between different working conditions (job demands) and individuals’ mental health. Specifically, working more than forty hours per week, bad schedule adaptation to social and family commitments and suffering physical violence from the host children were associated with marked risk increments for depressive symptoms.

The observed high prevalence of depressive symptoms is in consonance with the 20% prevalence of depression among migrant workers in Europe observed in a large-scale review including data from all EU member states [31]. These results coincide, as well, with the 29% prevalence of depressive disorders among Turkish migrants [32], and 45% prevalence of distress among Latin American migrants in Germany [9] reported in previous studies.

In addition, previous studies have shown that au pairs often suffer from poor working conditions. For instance, Sollund and colleagues reported that au pairs in Norway normally exceeded the legal working hours [1], while another study reported that one out of three au pairs work more than 30 h per week [33]. Moreover, an investigation from Ireland observed that 26% of au pairs worked between 40 and 60 h per week [34]. Furthermore, in a study by O’Connor et al., poor mental health, including depressive symptoms, among Latin American migrants were associated with poor working conditions, violence at work and a low level of education, among other factors [35]. Similar results have also been yielded by Vahabi et al., where overwork was related to a higher prevalence of depression among this population [36]. Furthermore, Carlos and Wilson mentioned that 67% of live-in caregivers (a population comparable with au pairs), presented poor mental health due to overload and overtime work [37,38]. Moreover, the above-mentioned Irish study revealed that 21% of au pairs in that study did not receive regular breaks, 15% had to be ‘on call’ at night, 27% worked on Sundays and 30% reported not getting any holidays [34]. Therefore, 36% of au pairs faced stress and 51% claimed the situation was worse than they expected due to overwork [34]. All these conditions, such as lack of privacy, individuals’ powerlessness to have control over their living–working conditions and overwork increased poor mental health and the prevalence of depression among au pairs [36]. In total, all these results support the findings of the present study.

Furthermore, physical violence by the host children was associated with depressive symptoms among Spanish-speaking au pairs in this study. This finding is in line with another study reporting that live-in caregivers suffered stress due to lack of social and family support, disobedience from children, work overload, overtime and lack of permanent residency status, among other factors [39]. Furthermore, an ethnographic research paper in England reported that some au pairs suffered physical violence at the hands of the host children because they did not accept the au pair as caretaker, which might contribute to stress [40].

Regarding the longitudinal assessment, depressive symptoms did not show a noteworthy change over time. Based on the comparison of the prevalence of MDS of the participants who dropped out during the follow-ups (12.2%) and those that remained (3.1%), we assume that au pairs who faced poor working conditions and developed depressive symptoms might have returned to their country of origin before finishing the au pair year (the so-called healthy migrant effect) [16,41]. This assumption matches observations from Hondagneu-Sotelo who described that Latin American au pairs find it difficult or risky to express their concerns due to cultural differences [42]. Hence, host families and au pairs avoid discussions about difficulties and problematic situations and decide instead to terminate the au pair contract [42]. Moreover, in an earlier cross-sectional study, experienced Spanish-speaking au pairs presented more than two times the prevalence of MDS and almost two times the prevalence of depressive syndromes than the newcomers [15]. Therefore, it can be assumed that the present results might represent an underestimation of the true degree of depressive symptoms among au pairs.

The strengths of this study include the longitudinal approach that evaluated how the symptoms of depression faced by the au pairs changed over the time. Secondly, the main inclusion criteria of being a “newcomer au pair” allowed us to determine the pre-existing depressive symptoms among the participants and to identify the variation in their symptoms over time. Third, the internet-based sampling method (conventional and Facebook snowball sampling) helped to reach a dispersed and vulnerable population (e.g., migrants), and to increase the geographical scope and sample size [43]. According to Baltar and Brunet, sampling via social media is more successful (77% response) than conventional snowball (42% response) [44]. Fourth, incentives (shopping vouchers worth five euros) ensured the fulfilment of the online surveys and prevented dropouts during the follow-ups [45]. Fifth, the application of the Spanish version of the internationally standardized instruments allowed the comparison of this study with other international studies [46]. Finally, the authors chose the GEE model to analyze the within-subject correlations over time [25]. This is one of the most suitable methods for longitudinal studies with discrete outcomes [26]. Furthermore, GEE includes a semi-parametric regression-based strategy for handling missing data, thus avoiding the imputation data process [28].

This study, nevertheless, presents some limitations. The non-random sampling method did not allow the calculation of a response rate and the representativeness of the study population could not be estimated. According to Rothman et al., findings are distinguished by the empirical goal of understanding a phenomenon (longitudinal studies) and the practical goal of applying this information to a specific population (cross-sectional studies) [47]. To rely on descriptive results as in cross-sectional studies, researchers should have a representative sample. However, these descriptive results cannot justify how a human behavioral phenomenon works. Understanding a phenomenon is based solely on controlled comparisons, not population representativeness. Therefore, in this study, we aimed to identify potential risk factors associated with depressive symptoms over time, rather than focusing only on describing the study population and their generalization.

Another limitation of this study was the selection of exclusively Spanish-speaking au pairs as study population. Therefore, the results may not be transferable to au pairs from other countries. Finally, as this study took a quantitative approach, we recommend further qualitative research to identify moods, feelings and possible indicators of depressive symptoms that were not included in this study.

Based on the 80 percent loss of participants with MDS between two and six months of stay in Germany, we assume that the initial two months of the au pair program are the most critical for developing depressive symptoms. Therefore, we recommend that au pair organizations carry out interventions before and during the second month. Interventions with the host families should clarify the roles, obligations and duties of au pairs, as well as the mutual benefits acquired. Authorities and au pair organizations may monitor whether au pairs are working a maximum 30 h per week, are given adequate time to attend language classes and have at least one full free day per week [48]. Finally, especially for au pairs who are not linked to any organization, we recommend psychological counseling and focus groups led by authorities as well as experienced au pairs. This may create a safe atmosphere for au pairs outside of their workplace (host family) and prevent poor mental health.

## 5. Conclusions

As we hypothesized, overwork, bad schedule adaptation to social and family commitments and physical violence from the host children were identified as potential risk factors for depressive symptoms among Spanish-speaking au pairs living in Germany. This scenario contradicts the definition of au pair, which itself means ‘at par,’ ‘at equal shares’ or ‘on mutual terms.’ However, these associations did not change over time. Most of the au pairs who presented depressive symptoms dropped out the study in the early phase. We assume that they returned to their home countries.

This knowledge could be of interest for future au pairs as well as for policy makers, au pair agencies and host families. Together, they could improve awareness and monitoring of working conditions, implement intervention strategies and ensure appropriate guidance before and during the au pair program.

Finally, we expect that this study will provide valuable information for au pair agencies to support their participants before they go to the host country, to prevent poor mental health and to create “fair-work” au pair programs.

## Figures and Tables

**Figure 1 ijerph-18-06940-f001:**
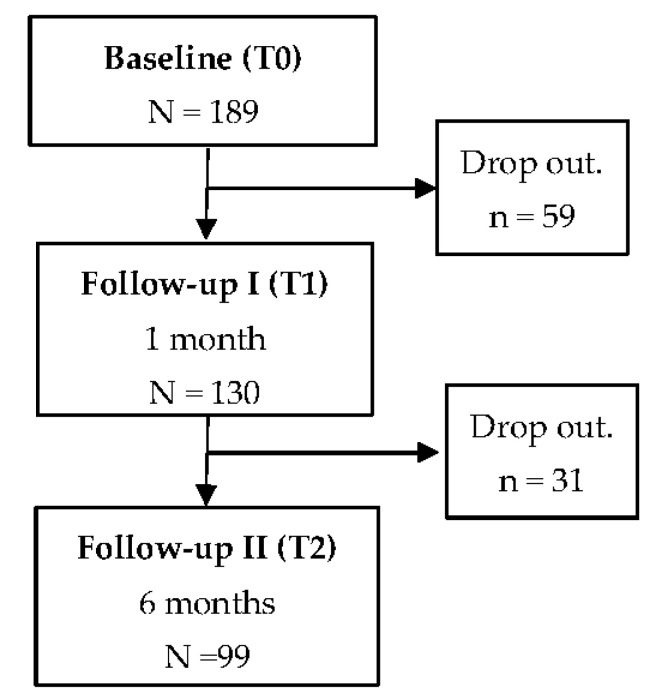
Number of participants in each stage of the study.

**Table 1 ijerph-18-06940-t001:** Descriptive data of participants who remained in the study through all three time points and the participants who dropped out of the study.

Characteristics		Participants	Dropouts	*p*-Value
		*N* = 99	*N* = 90	
		*n* (%)	*n* (%)	
**Gender**	Female	87 (89.7)	82 (89.1)	0.81
**Age (years)**	18–21	30 (30.9)	43 (47.3)	0.03
	22–24	44 (45.4)	25 (28.6)	
	25–28	23 (23.7)	22 (24.2)	
**Higher education**	Yes	80 (86.0)	68 (75.6)	0.09
**Region of origin**	Spain	10 (10.3)	8 (8.7)	0.54
	Colombia	53 (54.6)	40 (43.5)	
	Mexico and Central America	22 (22.7)	29 (31.5)	
	South America (w/o Colombia)	11 (11.3)	13 (14.1)	
**Settlement of residence ***	Towns	36 (38.3)	13 (41.2)	0.84
	Minor city	14 (14.9)	4 (17.6)	
	Major city	44 (46.8)	13 (41.2)	
**Region of residence in Germany ***	Northern	19 (20.5)	5 (12.5)	0.46
	Southern	33 (36.4)	10 (37.5)	
	Eastern	9 (9.1)	5 (12.5)	
	Western	31 (34.1)	10 (37.5)	
**Working hours per week ***	>40 h	16 (16.8)	4 (12.1)	0.59
**Extra hours of babysitting *^,^** **^#^**	Yes	62 (70.3)	21 (70.0)	0.82
**Working on holidays ***	Yes	31 (35.2)	11 (35.0)	0.87
**Days off per week ***	One day	26 (27.7)	11 (35.3)	0.37
	Two days	68 (72.3)	19 (64.7)	
**Schedule’s adaptation to social & family commitments ***	Well	71 (80.7)	26 (85.5)	0.99
**Au pair agency contract ***	Yes	46 (52.3)	15 (50.0)	0.69
**Additional job besides au pair ***	Yes	10 (11.4)	5 (12.5)	0.53
**Violence at work ***	Yes	8 (5.7)	1 (2.5)	0.55
**Physical violence by the host children ***	Yes	26 (29.9)	10 (32.5)	0.73
**Verbal offenses ***	Yes	20 (23.0)	10 (33.0)	0.50
**Depressive symptoms**	DS-	73 (73.4)	64 (71.1)	0.07
	DS+	15 (15.3)	8 (8.9)	
	ODS	8 (8.2)	7 (7.8)	
	MDS	3 (3.1)	11 (12.2)	

T0 = Baseline measure; T1 = 1-month follow-up; T2 = 6-month follow-up. Participants: those who remained in the study through all three time points. Dropouts: those who dropped out of the study during the follow-ups. * Variables assessed from T1 to T2 (dropouts *N* = 31). DS-: no reported depressive symptoms. DS+: at least one of the required screening symptoms is fulfilled, but the total symptom score is below the threshold diagnosis. ODS: Other Depressive Syndrome: 2–4 reported depressive symptoms and one of the symptoms is depressed mood or anhedonia. MDS: Major Depressive Syndrome: ≥5 reported depressive symptoms and one of the symptoms is depressed mood or anhedonia. ^#^ Taking care of children while parents are away from home due to leisure activities.

**Table 2 ijerph-18-06940-t002:** Prevalence of Depressive Symptoms (PHQ-9) among Spanish-speaking au pairs living in Germany at T2 = 6-month follow-up (N = 99) by potential risk factors.

Characteristics		Depressive Symptoms ^#^	*p*-Value
		*n* (%)	
**Gender**	Female	24 (27.3)	0.99
	Male	2 (25.0)	
**Age (years)**	18–21	8 (26.7)	0.80
	22–24	13 (29.5)	
	25–28	5 (20.8)	
**Higher education**	No	2 (15.4)	0.51
	Yes	21 (25.9)	
**Region of origin**	Spain	4 (40.0)	0.22
	Colombia	17 (32.1)	
	Mexico and Central America	4 (18.2)	
	South America(w/o Colombia)	1 (8.3)	
**Settlement of residence**	Towns	11 (28.9)	0.81
	Cities	15 (25.0)	
**Region of residence in Germany**	Northern	4 (20.0)	0.85
	Southern	10 (28.6)	
	Eastern	3 (33.3)	
	Western	9 (26.5)	
**Working hours per week**	≤40 h	18 (22.8)	0.05
	>40 h	8 (44.4)	
**Extra hours of babysitting ***	No	4 (19.0)	0.57
	Yes	22 (28.6)	
**Working on holidays**	No	13 (19.1)	0.02
	Yes	13 (43.3)	
**Days off per week**	One day	9 (33.3)	0.44
	Two days	17 (23.9)	
**Schedule’s adaptation to social & family commitments**	Bad	13 (56.5)	0.01
	Well	13 (17.3)	
**Au pair agency contract**	No	12 (25.0)	0.82
	Yes	14 (28.0)	
**Additional job besides au pair**	No	23 (26.4)	0.99
	Yes	3 (27.3)	
**Violence at work**	No	22 (24.4)	0.02
	Yes	3 (100.0)	
**Physical violence by the host children**	No	11 (16.9)	0.01
	Yes	14 (51.9)	
**Verbal offenses**	No	14 (20.3)	0.02
	Yes	9 (45.0)	

*p* value calculated with Fisher’s exact test. * Taking care of children while parents are away from home due to leisure activities. ^#^ Depressive Symptoms: includes DS+, ODS, and MDS.

**Table 3 ijerph-18-06940-t003:** Statistical analysis using the McNemar test between the dependent variable depressive symptoms in T0 and T1 with T2 follow-ups (*N* = 99).

Depressive Symptoms ^#^		T2		*p*-Value
		No	Yes	
		*n* (%)	*n* (%)	
**T0**	**No**	61 (82.4)	13 (17.6)	0.83
	**Yes**	11 (45.8)	13 (54.2)	
**T1**	**No**	61 (83.6)	12 (16.4)	0.99
	**Yes**	11 (44.0)	14 (56.0)	

*p* value calculated with McNemar. T0 = Baseline measure; T1 = 1-month follow-up; T2 = 6-month follow-up. ^#^ Depressive symptoms: includes DS+, ODS and MDS.

**Table 4 ijerph-18-06940-t004:** Generalized estimating equations (GEEs) for depressive symptoms (*N* = 189).

Characteristics		Crude OR	Adjusted OR
		(95% CI)	(95% CI)
**Working hours per week**	≤40 h	1	1
	>40 h	2.88 (1.32–6.28)	3.47 (1.46–8.28)
**Extra hours of babysitting ***	No	1	N/A
	Yes	1.88 (0.32–11.09)	
**Working on holidays**	No	1	1
	Yes	2.66 (1.25–5.66)	1.50 (0.71–3.18)
**Days off per week**	One day	1	N/A
	Two days	0.67 (0.30–1.49)	
**Schedule’s adaptation to social & family commitments**	Good	1	1
	Bad	1.31 (1.34–8.20)	2.24 (0.95–5.28)
**Au pair agency contract**	No	1	N/A
	Yes	1.70 (0.71–4.05)	
**Additional job besides au pair**	No	1	N/A
	Yes	0.68 (0.23–2.02)	
**Violence at work**	No	1	N/A
	Yes	0.78 (0.13 −4.49)	
**Physical violence by the host children**	No	1	1
	Yes	5.34 (2.33–12.21)	4.95 (2.16–9.75)
**Verbal offenses**	No	1	1
	Yes	4.38 (1.91–10.06)	1.63 (0.66–4.03)
**Follow-up time** **^+^**	T0	1	1
	T1	1.08 (0.73–1.58)	1.05 (0.53–1.50)
	T2	0.99 (0.61–1.61)	0.81 (0.42–1.58)

* Taking care of children while parents are away from home due to leisure activities. OR: odds ratio; CI: 95% confidence interval. Adjusted for working hours per week, working on holidays, schedule’s adaptation, physical violence, verbal offenses and follow-up times. ^+^ Follow-up times: T0 = Baseline measure; T1 = 1-month; T2 = 6-month follow-up.

## Data Availability

The data presented in this study are available on request from the corresponding author. The data are not publicly available due to data protection (Art. 26 BayDSG).

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
