# Peer review of "Working Conditions as Risk Factors for Depressive Symptoms among Spanish-Speaking Au Pairs Living in Germany—Longitudinal Study"

_ijerph, 2021, doi:10.3390/ijerph18136940_

Round 1
Reviewer 1 Report
Dear Author,
Many thanks for the opportunity to review your manuscript. The research is of great interest for uncovering this group of young 'workers' whose working conditions and mental health problems could have drawn more attention. Potentially the paper could make significant contributions, but to achieve this the paper may need a number of improvements:
1. The definition of 'depressive symptoms' should be given as this is one of the key terms/variables of the study. You mentioned the use of 9 depressive symptom criteria but these 9 items could have been included in the paper as texts, notes or a table.
2. Could you make a further discussion about how the sociodemographic characteristics are linked with depressive symptoms?
3. What are the research questions, or hypotheses? In both cases, you may want to discuss whether your findings can answer the question or support the hypotheses.
4. It seems the reason for undertaking such study includes (1) the lack of longitudinal research (2) follow-up of a previous research by yourself. This is OK, but a stronger reason could be to address the significant gap left by the existing findings for au pairs in Germany - could be a lack of understanding of a relationship between the symptoms and some factors, could be the inadequate attention paid to Spanish au-pairs in northern Europe, or a void on public health practice. If you could address these issues in the introduction, and state that these were the reasons for the significance of the study, it will help the paper to justify its positioning.
5. It is less clear how the study contributes to the field of study. Despite a number of findings aligned with existing literature, the paper has not made it clear about the extent to which its findings can help advance our understanding of certain knowledge in an academic field. Nor has the paper (in the 'Discussion' part) explicitly acknowledged its theoretical contribution - how would the findings add on existing theories that explain the factors influence depressive symptoms, and how the new findings help develop novel angle or perspective on this matter? This is perhaps the biggest weakness of the paper, despite its relatively detailed examination how the findings can be supported by existing literature.
6. Related, you might want to highlight the overarching theory used (a theoretical framework) to lead to the research question/hypotheses, and analyse your findings. At the moment, this seems missing.
Reviewer 2 Report
Thank you for your submission. I believe the topic is very interesting and contribute significantly to support the au-pairs working in foreign countries.
There are few points to be considered revision.
[Minor Revision]
- Since the strength of this study is the longitudinal data and the main outcome are depressive syndromes, working conditions, violence at the workplace, and socio-demographic characteristics at 3 time point, you may wish to add a table on PHQ-9 and other results of all 3 times of survey (T0, 1 and 2) in the result section.
- All tables are difficult to follow. You could make sure that all the numbers are aligned and not over wrapped.
Round 2
Reviewer 1 Report
Dear Author
Thanks for the revised manuscript. Based on the new version where you have inserted new texts (red highlights), you did not seem to address the following issues raised in the previous round of review. They are (in original numbering and texts):
3. What are the research questions, or hypotheses? In both cases, you may want to discuss whether your findings can answer the question or support the hypotheses.
4. It seems the reason for undertaking such study includes (1) the lack of longitudinal research (2) follow-up of a previous research by yourself. This is OK, but a stronger reason could be to address the significant gap left by the existing findings for au pairs in Germany - could be a lack of understanding of a relationship between the symptoms and some factors, could be the inadequate attention paid to Spanish au-pairs in northern Europe, or a void on public health practice. If you could address these issues in the introduction, and state that these were the reasons for the significance of the study, it will help the paper to justify its positioning.
5. It is less clear how the study contributes to the field of study. Despite a number of findings aligned with existing literature, the paper has not made it clear about the extent to which its findings can help advance our understanding of certain knowledge in an academic field. Nor has the paper (in the 'Discussion' part) explicitly acknowledged its theoretical contribution - how would the findings add on existing theories that explain the factors influence depressive symptoms, and how the new findings help develop novel angle or perspective on this matter? This is perhaps the biggest weakness of the paper, despite its relatively detailed examination how the findings can be supported by existing literature.
6. Related, you might want to highlight the overarching theory used (a theoretical framework) to lead to the research question/hypotheses, and analyse your findings. At the moment, this seems missing.
Best wishes
